# Design, Characterization, and Performance of Woven Fabric Electrodes for Electrocardiogram Signal Monitoring

**DOI:** 10.3390/s22155472

**Published:** 2022-07-22

**Authors:** Meiling Zhang, Ningting Guo, Qian Gao, Hongqiang Li, Zhangang Wang

**Affiliations:** 1School of Textile Science and Engineering, Tiangong University, Tianjin 300387, China; zhangmeiling@tiangong.edu.cn (M.Z.); 2031010129@tiangong.edu.cn (N.G.); gggaoqian@163.com (Q.G.); 2School of Electronic and Information Engineering, Tiangong University, Tianjin 300387, China; lihongqiang@tiangong.edu.cn; 3School of Software, Tiangong University, Tianjin 300387, China

**Keywords:** fabric electrode, fabric weave, ECG signal, weft density

## Abstract

Conductive gel needs to be applied between the skin and standard medical electrodes when monitoring electrocardiogram (ECG) signals, but this can cause skin irritation, particularly during long-term monitoring. Fabric electrodes are flexible, breathable, and capable of sensing ECG signals without conductive gel. The objective of this study was to design and fabricate a circular fabric electrode using weaving technology. To optimize the woven fabric electrode, electrodes of different diameter, fabric weave, and weft density were devised, and the AC impedance, open-circuit voltage, and static ECG signal were measured and comprehensively evaluated. Diameter of 4 cm, 12/5 sateen, and weft density of 46 picks/cm were concluded as the appropriate parameters of the fabric electrode. ECG signals in swinging, squatting, and rotating states were compared between the woven fabric electrode and the standard medical electrode. The results showed that the characteristic waveform of the woven fabric electrode with 86.7% improved data was more obvious than that of the standard medical electrode. This work provides reference data that will be helpful for commercializing the integration of fabric electrodes into smart textiles.

## 1. Introduction

Currently, cardiovascular diseases are one of the major factors affecting human health, and lack of timely resuscitation after cardiac events is a common problem [1]. Analysis and alerting using real-time electrocardiogram (ECG) signals could offer a solution to decrease the death toll caused by cardiovascular diseases [2,3]. Due to the limitations of large volume and power supply conditions, the monitors of conventional ECG tests can only be used in hospitals and specific sites. The Holter ECG recorders can record patients’ heart activity at rest, during activity, on eating, when learning and in various other conditions over a long time and for 24 h continuously. However, 5–7 electrodes are required to be attached to the chest, which is uncomfortable [4].

In a conventional ECG measuring system, ECG signals are recorded by adhering silver/silver chloride (Ag/AgCl) electrodes to the skin to capture bio-potential signals, and these signals are converted into electrical signals by a digital filtering system [5]. Ag/AgCl electrodes are normally used with a conductive gel and an adhesive layer to reduce the impedance between the electrode and the skin. However, long-term use on the human skin may cause skin irritation, allergies, or discomfort [6,7], while the drying of the conductive gel over time leads to low quality of signals and larger impedance of the electrode-skin interface [8,9], which prevents long-term monitoring of the patient [10]. Fabric electrodes have a textile structure and can sense bioelectric signals from the surface of the skin [11,12]. They are flexible and breathable, so they are more comfortable than conventional metal plate electrodes for long-term monitoring. In addition, as they can be easily integrated into a garment, the conductive gel and adhesives can be dealt with by adjusting suitable pressure between the skin and fabric electrodes, so they are skin-friendly and can achieve long-term and real-time monitoring of ECG signals.

Fabric electrodes are usually made by weaving, knitting, or embroidering with conductive yarns; or by coating or printing conductive polymers onto non-conductive fabrics. They must be in contact with the skin to acquire signals, and their structure may affect the interface between the electrodes and the skin. Knitted fabrics are easily deformed and experience large changes in resistivity when subjected to pressure, which will make the ECG signal unstable [13]. Woven fabrics have suitable porosity, stable structure, and good electrochemical stability [14]. Unfortunately, for fabric electrodes with a plain weave, the resistance changes are unstable and the acquired signal has a large error when the electrode is subjected to different pressures [12]. Fabric electrodes with a honeycomb weave [15] show relatively stable resistance changes when subjected to different pressures, but the intensity of the collected signals is weak due to their many perforations and smaller contact area with the skin. Compared with plain fabrics, the 5/3 satin weave has the advantages of superior comfort and lower skin–electrode impedance [16] with the same materials, warp, and weft density, so it is more suitable for preparing fabric electrodes. However, to the best of our knowledge, little systematic studies on sateen weave have been conducted. A study [17] reported the effect of 1/15 sateen, 1/15 twill, 1/15 broken twill and 1/15 birds-eye structure on electrode performance, but there are no series of studies on sateen weave. Regarding the shape of fabric electrodes, most studies have focused on square electrodes [18,19] as opposed to circular electrodes. In addition, few studies have investigated how weft density affects the performance of fabric electrodes for monitoring signals. Other conductive electrodes made with deposition [20,21], coating [22,23,24,25,26], and printing [27,28] have also been studied. However, one drawback of coated fabrics is that the conductive layer will be gradually abraded over time, resulting in larger resistance and impedance of the electrodes, which will make the acquired ECG signal more inaccurate.

In this study, we describe the fabrication of fabric electrodes for ECG monitoring by weaving with silver-plated nylon filament. Electrodes with different diameter, fabric weave, and weft density were systematically investigated to determine the optimized parameters for which combination of AC impedance, the open circuit voltage, and the static ECG signal. In order to verify the performance of the optimized fabric electrodes, the dynamic ECG signals acquired by optimized fabric electrodes and standard medical electrodes were compared. The proposed fabrication approach will help pave the way for the development of fabric electrodes in wearable monitoring garments.

## 2. Materials and Methods

### 2.1. Materials

The warp yarn was 30S/2 viscose/nylon (Dongguan Zhengyu Textile Co., Ltd., Dongguan, China). The weft yarn was 140D plied yarn by silver-plated nylon filament (Qingdao Hengtong Weiye Special Fabric Technology Co., Ltd., Qingdao, China), selected for its electrical conductivity, physiological compatibility, and antibacterial activity. The fabric electrode does not require conductive gel or adhesive. Sodium chloride solution (0.9%) was used to measure the electrochemical properties.

### 2.2. Fabric Electrode Design and Fabrication

Conductive weft yarn is critical for the design of the woven fabric electrode because the side of the fabric on which the conductive weft yarns float more, will be the side in contact with the skin. Therefore, sateen weave was selected to reduce the AC impedance and improve the accuracy of the collected ECG signal.

The effects of three factors—diameter, fabric weave, and weft density (the number of weft yarns per unit length)—on the fabric electrodes were investigated with the single variable method, meaning that the optimized values determined in previous tests were applied to subsequent tests. The parameters of the three factors are shown in Table 1.

Experiments were performed based on diameters of 1 cm, 2 cm, 3 cm, 4 cm, and 5 cm, fabric weave of 5/2 sateen and weft density of 35 picks/cm. The fabric electrode of a certain diameter can acquire a better effect, this diameter is defined as the “Optimized Diameter”. The definition of optimized fabric weave is the same as that of optimized diameter.

The fabric electrode was woven by dobby loom (Tianjin Jiacheng Electromechanical Equipment Co., Ltd., DWL5016, Tianjin, China), as seen in Figure 1a, where the white yarn is warp yarn, the brown is weft yarn. The circular woven electrode is cut out from the whole woven fabric (Figure 1b). The shape of the fabric electrode makes reference to the circular shape of the medical electrode.

The weave diagram of 5/2 sateen is shown in the inset of Figure 1c. The intersection of warp and weft yarns is the interlacing point, when the warp yarn floats on top of the weft yarn, it is called the warp interlacing point, as the black point in the inset of Figure 1c. When the weft yarn floats on top of the warp yarn, it is called the weft interlacing point. Four weft-interlacing points connected together are called weft floating, as the white points in the inset of Figure 1c.

The conductive yarns are woven as weft yarns adopting sateen which means that the front side of the fabric is mostly conductive weft floating (Figure 1c). Therefore, long conductive weft floating can be shown in the front side which will contact the skin to transmit the ECG signal.

### 2.3. Assembly and Principle of Fabric Electrodes

Figure 2a outlines the assembly structure of the fabric electrode. Foam plastics were placed between the elastic bandage and the fabric electrode to improve the contact area between electrode and skin and maintain uniform force on the electrode. The fabric electrodes are combined on an elastic bandage with Velcro. The bandage was mounted on the body to measure the signal in Figure 2b. The Velcro position can be adjusted to meet with the requirement of consistent pressure based on the tested human body. The ECG signal acquisition setup is depicted in Figure 2c. The ECG signal collected with the assembled fabric electrode was processed by the ECG acquisition board and then fed back to the PC to record the voltage amplitude of the ECG signal.

The skin surface and the fabric electrode each act as a conductive flat plate, and the ECG signals are coupled from the human skin to the fabric electrode by direct current coupling. The signals from the two electrodes are directly passed to the differential amplifier as shown in Figure 2d. Because the ECG signal gathered by the fabric electrode is slightly weak, it must be processed with differential operation through the signal amplifier to obtain a stronger ECG signal. ECG monitoring equipment developed by Keil5 is composed of a microcontroller (Master control MCU) and an ECG analog front end. The ECG signal collected by the analog front end is transmitted to the PC through the serial port (UART) after the data format processing by the microcontroller. The PC uses MATLAB algorithm to denoise the ECG signal and extract the characteristic value.

### 2.4. Electrochemistry and Human Subject Testing

The CHI660E electrochemical workstation (Shanghai Chenhua Instruments Co., Ltd., Shanghai, China) was used to measure AC impedance and open-circuit voltage, 0.9% sodium chloride solution was selected as the electrolyte solution to create a conductive environment. The positively charged cations and negatively charged anions dissociated by the electrolyte move towards the corresponding electrode under the action of the external electric field resulting in the directional movement of electrons to conduct electricity. The surface of the fabric on which the conductive weft yarns floated more was taken as the working side of the electrode. The initial voltage was set to 10 mV when testing AC impedance, and the scanning frequency range was 0.01–1000 Hz.

The ECG signal was measured at a room temperature of 25 ± 1 °C and a relative humidity of 65 ± 2%. The subject was required to sit for 5 min while breathing evenly before the static ECG signal was measured. For dynamic ECG testing, three dynamic conditions in the standing position were devised: swinging (arms swing back and forth with a range of motion from 0° to 90°), squatting (arms hang down naturally while the subject slowly squats until they are completely squatting), and rotating (arms hang down naturally, rotating 360° in place). These three motions were chosen with reference [28] taking into account the common actions of daily life. A unipolar chest lead was used, the center points of the positive and negative electrodes were 8–9 cm apart, and the test position was 3–4 cm below the thoracic nipple. It was confirmed that the complete characteristic waveform could be obtained from the recorded ECG signals.

To assess the ECG signal of the woven fabric electrodes, tests were performed with five healthy males with no heart problem history. The ECG waveform of subject 1 was listed in this text for analysis. The collected signals of all subjects were compared based on the amplitudes of the characteristic peaks. Signal-to-noise ratio (SNR) refers to the ratio of ECG signal to noise (irregular additional signal that does not exist in the ECG signal) in an electronic device or electronic system. The larger the SNR, the better the signal quality. The SNR is calculated using MATLAB software with the following Equation (1):(1)SNR=20log (Vs/Vn) 
where *Vs* represents the valid values of the signal and *Vn* represents the valid values of the noise voltage.

## 3. Results and Discussion

The AC impedance and open-circuit voltage were measured by the electrochemical workstation and the ECG signals in the static state were recorded. In addition, the ECG signals in the squatting, rotating, and swinging states were recorded and the SNR was calculated for each state.

### 3.1. Fabric Electrode Diameter

Based on standard medical electrodes, the AC impedance does not exceed 3 kΩ at 10 Hz, the disturbance current does not exceed 100 μA, and the open-circuit voltage is generally kept under 100 mV.

Figure 3 describes the electrochemical performance of fabric electrodes (5/2 sateen, weft density 35 picks/cm) with different diameters. As shown in Figure 3a, the larger the area of the fabric electrodes, the smaller the AC impedance at the same frequency for fabric electrodes of different diameters.

The equivalent circuit of AC impedance for a pair of fabric electrodes is shown in Figure 4. The fabric electrode can be modeled as a parallel circuit described by an equivalent AC impedance (*Z*), which is composed of the resistance of the fabric electrode (*R_t_*) and the electric double-layer capacitor between the fabric electrode and the sodium chloride solution (*C_t_*). The resistance of the sodium chloride solution (R_e_) is negligible.

Equation (5) is obtained based on Equations (2)–(4), which provide the expressions for resistance, capacitance, and AC impedance.
(2)$$R_{t}=\frac{\rho L}{S}$$
where *ρ* is the electrical resistivity of the material, *S* is the skin–electrode contact area, and *L* is the thickness of the fabric electrode, which is a constant value since the thickness remains the same for all the fabric electrodes.
(3)$$C_{t}=\frac{\varepsilon S}{4\pi kd}$$
where *ε* is the permittivity of the dielectric layer, *k* is the electrostatic force constant, *ω* is the angular frequency, and *d* is the radius of the negative ion (Cl^−1^) in the double capacitance:(4)$$\frac{1}{Z}=\frac{1}{R_{t}}+j\omega C_{t}$$
(5)$$\left|Z\right|=\sqrt{\frac{1}{{\left(\frac{1}{\rho L}\right)}^{2}+{\left(\frac{\omega \varepsilon }{4\pi kd}\right)}^{2}}}•\frac{1}{S}$$

According to Equation (5), theoretically, the increase of the effective contact area between the skin and electrode will result in a decrease of the AC impedance modulus |*Z*|. Therefore, increasing the diameter of the fabric electrodes can obviously reduce the impedance. The theoretical estimation indicated that the best signal would be achieved by the fabric electrodes with diameter of 5 cm. However, the signal transduction part will amplify the signal. With the diameter of 5 cm, the signal amplification will be excessive, which will lower the accuracy of the signal. Therefore, the fabric electrodes with a diameter of 4 cm had a better effect.

As can be seen in Figure 3b, the open-circuit voltages of all diameters were less than 100 mV, so only the amplitude and stability of the open-circuit voltage curve were considered when evaluating the electrodes. The amplitude of the 3 cm diameter electrode was 1.8 mV, the amplitudes of the 4 and 5 cm diameter electrodes were 3.9 mV, and the amplitudes of the others were over 5 mV. Therefore, the open-circuit voltages of the 3, 4, and 5 cm diameter electrodes were most stable. Excluding the 1, 2, and 5 cm diameter electrodes based on AC impedance and open-circuit voltage, the static ECG signals of the 3 and 4 cm diameter electrodes were analyzed.

As demonstrated in Figure 3c, the heart rate is 60 bpm in static state, five heartbeat cycles can be observed within 5 s. The expected heartbeat cycle can be measured from samples of 2, 3, and 4 cm diameter. The characteristic waveform of the 3 cm diameter electrode was obvious, but there were some noises. Ideally, it has no small convex peaks and should be smooth and straight; the waveform of the 4 cm diameter electrode was clear and stable with obvious QRS complex and T-wave, and it was smooth with almost no noise. As a result, 4 cm was confirmed as the optimal diameter for the electrodes.

### 3.2. Fabric Weave of Fabric Electrodes

Figure 5 illustrates the electrochemical properties of fabric electrodes (diameter 4 cm, weft density 35 picks/cm) with different fabric weaves. From Figure 5a, the lower the frequency, the higher the AC impedance, and the larger the float of the sateen weave, the lower the AC impedance.

According to Equation (4), it can be deduced that
(6)$$\left|Z\right|=\frac{R}{\sqrt{1+4\pi R_{t}{}^{2}f^{2}C_{t}{}^{2}}}$$
where *R_t_* is the resistance of the fabric electrode, *C_t_* is the electric double-layer capacitor between the fabric electrode and the sodium chloride solution, *Z* is the AC impedance, *f* is the frequency. It can be seen from the equation that the greater the frequency, the greater the AC impedance.

The floats of 5/2, 8/3, 10/3, 12/5, and 16/11 sateen were 2, 3, 3, 5, and 11, respectively. As the float of the sateen weave increased, the contact points between the fabric electrodes and the human skin increased, and the contact area increased, so that the AC impedance decreased according to Equation (5). The sateen weaves with the three largest floats were selected for further analysis.

The open-circuit voltages of electrodes with different fabric weaves were all less than 100 mV, as shown in Figure 5b. The 16/11 sateen was most stable, but its long floats meant the fabric could be easily snagged. The amplitude of 10/3 sateen was 10.8 mV and that of 12/5 sateen was 9.1 mV. Figure 5c demonstrates that the waveform of 10/3 sateen was stable, and the characteristic waveform could be observed, but the noise was large which can be seen in Figure 5c. The P-wave, QRS complex, and T-wave of 12/5 sateen could be clearly observed, and there was nearly no noise. All the samples of different density can observe the expected heartbeat cycle. Therefore, the best fabric weave for the electrodes was determined to be 12/5 sateen.

### 3.3. Fabric Electrode Weft Density

The electrochemical performance and static ECG signal of fabric electrodes (diameter 4 cm, 12/5 sateen) with different weft densities are provided in Figure 6. Figure 6a shows that the larger the weft density, the smaller the AC impedance.

The conductive weft yarns were regarded as parallel resistors. Because the area of the fabric electrode is fixed, the larger the number of yarns, the higher the weft density. On the basis of the electrical model, the resistance of the electrode is negatively related to the number of yarns, whereas the capacitance is positively related to the number of yarns. According to Equation (7) derived from Equation (4), the AC impedance diminished with the decrease of resistance and increase of capacitance. Therefore, the lowest AC impedance corresponded to the highest weft density of 46 picks/cm.
(7)$$\left|Z\right|=\sqrt{\frac{1}{{\left(\frac{1}{R_{t}}\right)}^{2}+\omega^{2}C_{t}{}^{2}}}.$$

As shown in Figure 6b, the amplitude of the fabric electrodes with larger weft densities of 24, 35, and 46 picks/cm was approximately 2 mV. As for their static ECG signals, presented in Figure 6c, the samples of 24, 35, and 46 picks/cm could observe the expected heartbeat cycle. The fabric electrodes with 24 picks/cm had a drift in the baseline and a lower R-wave. The fabric electrodes with 35 picks/cm had an inconspicuous characteristic waveform, a weaker signal, and a larger noise, as can be seen in Figure 6c. Meanwhile, the P-wave, QRS complex, and R-wave were clear and obvious, and there was almost no noise in the fabric electrodes with 46 picks/cm. Therefore, the weft density of 46 picks/cm was optimal.

In summary, the optimized circular fabric electrodes should have a diameter of 4 cm, fabric weave of 12/5 sateen, and weft density of 46 picks/cm.

### 3.4. Evaluation of Woven Fabric Electrodes in Dynamic States

#### 3.4.1. Measurement of Woven Fabric Electrodes in Dynamic States

The ECG signal and SNR of the woven fabric electrodes were compared with those of standard medical electrodes in three dynamic states. Figure 7, Figure 8 and Figure 9 show the ECG signals acquired from the woven fabric electrodes and standard medical electrodes during squatting, swinging, and rotating conditions.

It can be observed that the waveform baseline of the woven fabric electrodes had no obvious drift as shown in Figure 7b, and the signal strength of the QRS complex and R-wave was clearly visible with good signal stability as the amplitudes of the characteristic peaks remain basically the same in each heartbeat cycle, and there is almost no noise in the squatting state because the line has no small peak except for the characteristic peak.

The stability of the overall signal of the woven fabric electrodes was superior in the swinging state in Figure 8b, as there was lower drift in the waveform baseline. Additionally, the QRS complex was clear and obvious, and the signal strength of the R-wave was comparable with the medical electrodes, with almost no noise.

In the rotating state in Figure 9b, the waveform baseline had a slight drift, the QRS complex was obvious, and the signal intensity of the R-wave was strong for the woven fabric electrodes. The standard medical electrodes show obvious noise while the woven fabric electrodes exhibit a relatively stable, low-noise signal. The stable ECG signals obtained from the woven fabric electrodes reflect the excellent contact performance between the skin and the fabric electrodes.

The SNR data were calculated using MATLAB and are presented in Table 2. The SNR of the woven fabric electrodes in each dynamic state was larger than that of the standard medical electrodes, with an increase of 18% in the swinging state, 7.8% in the squatting state, and 5.4% in the rotating state.

From the results of the dynamic ECG signal and SNR evaluation, it can be concluded that the woven fabric electrodes offer slightly better performance than the standard medical electrode and can meet the same monitoring requirements as the standard medical electrode.

#### 3.4.2. Efficacy and Reliability of Woven Fabric Electrodes

ECG signals of five healthy males were analyzed to evaluate the reliability and stability of the woven fabric electrode. The amplitudes of the P, R, and T characteristic peaks which were acquired using both fabric electrodes and medical electrodes were extracted and are listed in Table 3 in the squatting, swinging, and rotating states.

For the amplitude of the P-peak, 14 of the 15 data collected showed that the fabric electrode was larger than that of the medical electrode, so 93.3% of the fabric electrode data was better than that of the medical electrode.

For the R-wave, 9 of the 15 data collected from the fabric electrode were larger than that of the medical electrode, 2 of the 15 data showed little difference, so 73.3% of the fabric electrode data was greater than that of the medical electrodes.

As for the T-wave, 13 of the 15 data collected were larger, 1 of the 15 data showed little difference, so 93.3% of the fabric electrode data was superior.

In general, 39 of the 45 data collected were distinct which means 86.7% of the woven fabric electrode data is superior to that of medical electrode. Therefore, the woven fabric electrode is feasible and promising.

## 4. Conclusions

In this work, circular fabric electrodes were fabricated for ECG signal monitoring through a weaving technique. Different diameters, fabric weaves, and weft densities of the electrode were tested, and the optimized selections were found to be 4 cm, 12/5 sateen, and 46 picks/cm, respectively, based on the electrochemical properties and static ECG signals. In addition, the performance of the woven fabric electrodes was evaluated in swinging, squatting, and rotating states in terms of ECG signal and SNR and compared with the performance of standard medical electrodes in the same three states. The results showed that the characteristic waveform of the woven fabric electrode was more obvious, and the SNR was slightly larger than that of the standard medical electrode. Five different human subjects were employed to assess the performance of the electrodes, 86.7% of the fabric electrode data was superior to that for medical electrodes which demonstrates that the waveforms of the fabric electrode are better than those of the medical electrode. Moreover, with the fabric electrodes proposed in this study an important research foundation has been laid for the further development of fabric electrodes for ECG monitoring, and it would be a promising approach for application in wearable medical monitoring garments.

## Figures and Tables

**Figure 1 sensors-22-05472-f001:**
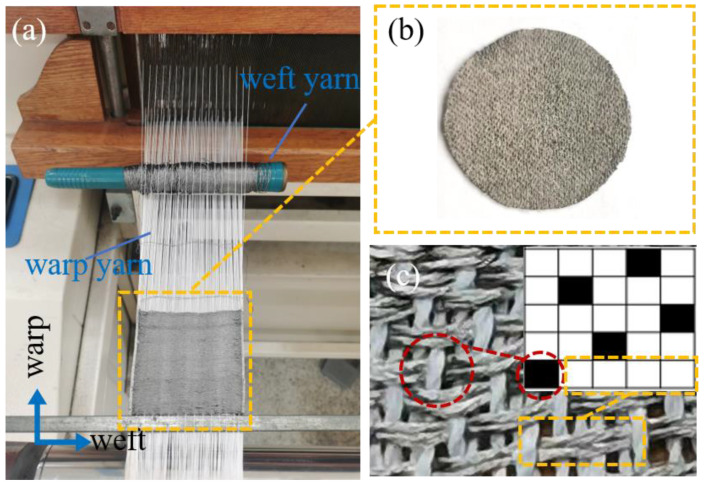
(**a**) The fabrication of the woven fabric in the loom. (**b**) The image of woven fabric electrode. (**c**) Weave diagram and physical diagram of 5/2 sateen (the front side).

**Figure 2 sensors-22-05472-f002:**
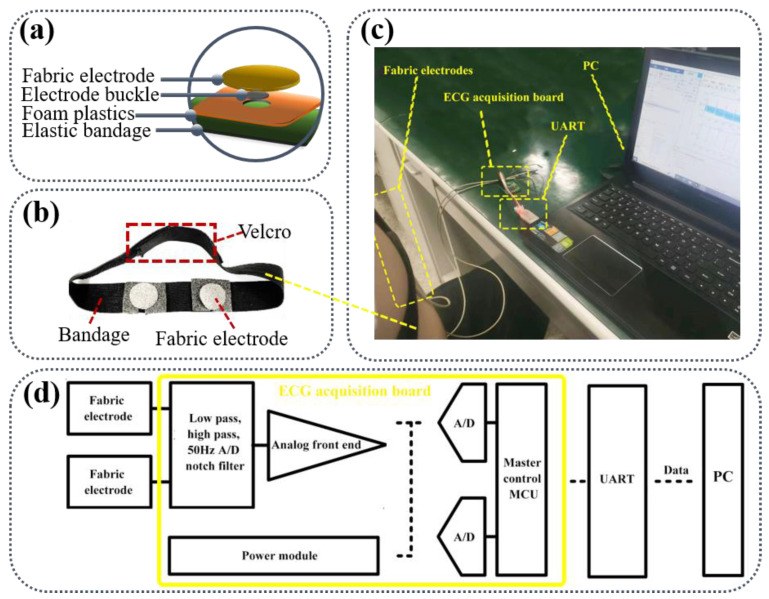
(**a**) The assembly structure of the fabric electrode; (**b**) The image of fabric electrodes on the bandage; (**c**) Schematic of ECG signal acquisition setup; (**d**) The development environment of ECG monitoring equipment.

**Figure 3 sensors-22-05472-f003:**
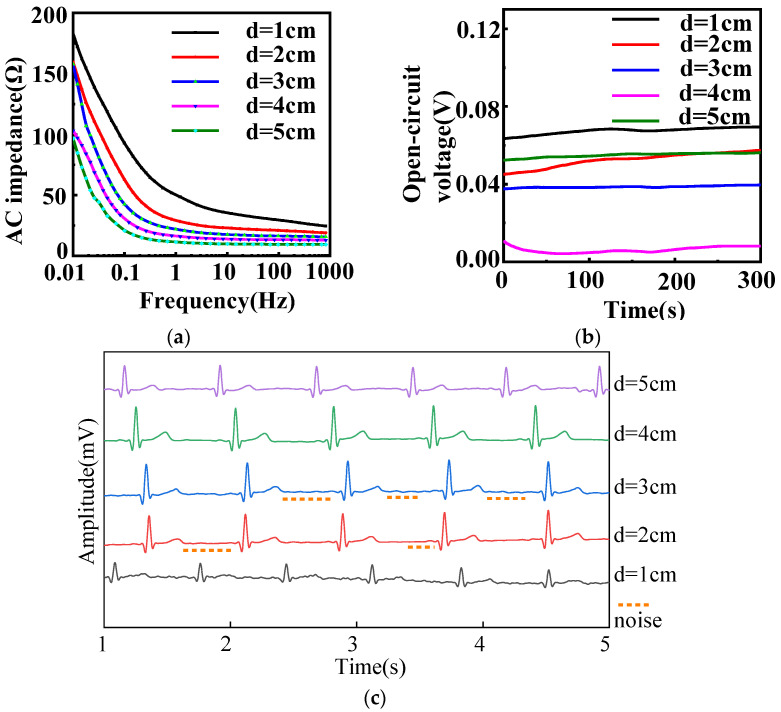
(**a**) AC impedance, (**b**) open-circuit voltage, and (**c**) static ECG signals of different diameters with 12/5 sateen and 35 picks/cm for fabric electrodes. (d represents the diameter).

**Figure 4 sensors-22-05472-f004:**
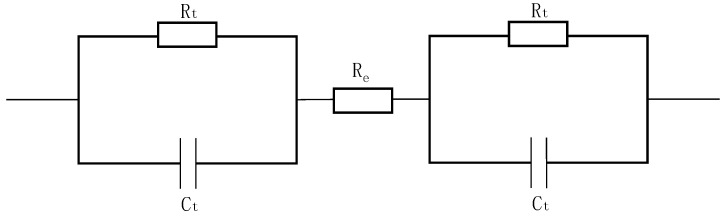
Equivalent circuit of fabric electrodes for testing AC impedance.

**Figure 5 sensors-22-05472-f005:**
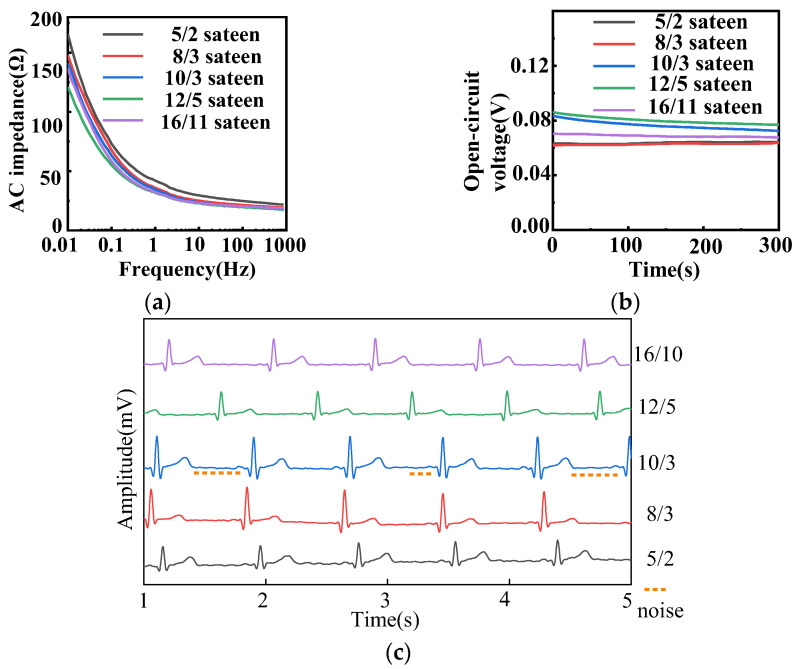
(**a**) AC impedance, (**b**) open-circuit voltage, and (**c**) static ECG signals of different fabric weaves with 4 cm diameter and 35 picks/cm for fabric electrodes.

**Figure 6 sensors-22-05472-f006:**
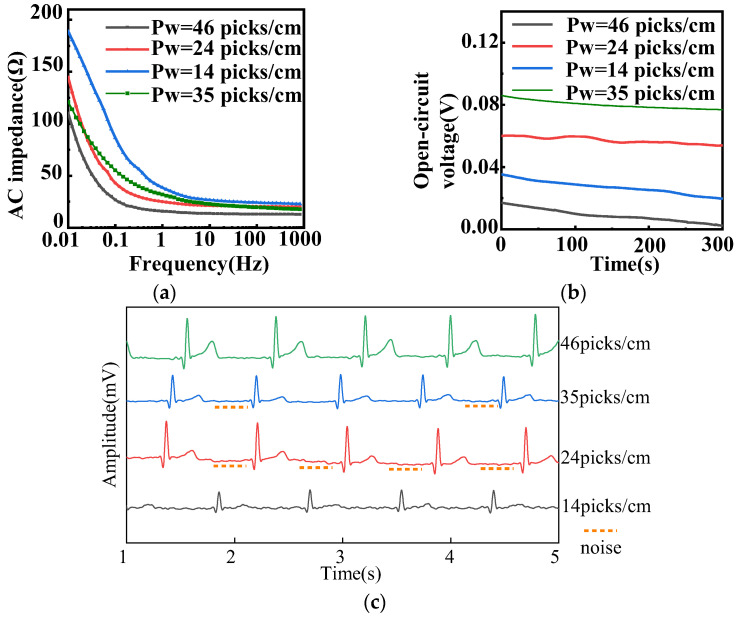
(**a**) AC impedance, (**b**) open-circuit voltage, and (**c**) static ECG signals of different weft densities with 4 cm diameter and 12/5 sateen for fabric electrodes. (Pw represents the weft density).

**Figure 7 sensors-22-05472-f007:**
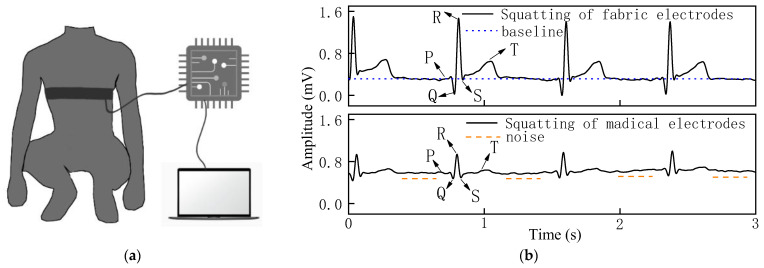
(**a**) Process of measuring ECG signal in squatting state. (**b**) ECG signal of woven fabric electrodes and standard medical electrodes in squatting state.

**Figure 8 sensors-22-05472-f008:**
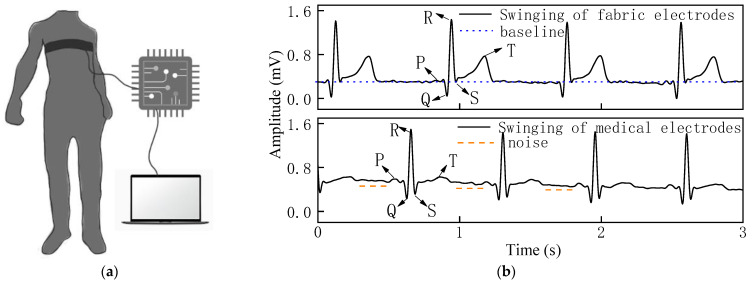
(**a**) Process of measuring ECG signal in swinging state. (**b**) ECG signal of woven fabric electrodes and standard medical electrodes in swinging state.

**Figure 9 sensors-22-05472-f009:**
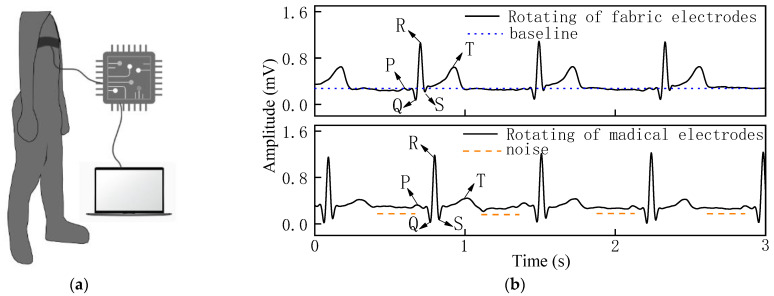
(**a**) Process of measuring ECG signal in rotating state. (**b**) ECG signal of woven fabric electrodes and standard medical electrodes in rotating state.

**Table 1 sensors-22-05472-t001:** Parameters of fabric electrode for each factor.

Factors	Proposed Parameters	Other Parameters	
Diameter	1 cm, 2 cm, 3 cm, 4 cm, 5 cm	5/2 sateen	Weft density 35 picks/cm
Fabric weave	5/2, 8/3, 10/3, 12/5, 16/11 sateen	Optimized diameter	Weft density 35 picks/cm
Weft density	14 picks/cm, 24 picks/cm, 35 picks/cm, 46 picks/cm	Optimized diameter	Optimized fabric weave

**Table 2 sensors-22-05472-t002:** SNR of the woven fabric electrodes and standard medical electrodes in various conditions.

Condition	Standard Electrodes/dB	Woven Fabric Electrodes/dB
Squatting state	42.4538	45.7301
Swinging state	35.4177	41.8488
Rotating state	42.7091	45.0049

**Table 3 sensors-22-05472-t003:** Comparison of amplitudes for P, R, and T characteristic peaks.

Subject	State	P (mV)		R (mV)		T (mV)	
		Fabric Electrode	Medical Electrode	Fabric Electrode	Medical Electrode	Fabric Electrode	Medical Electrode
1	Squatting	0.052 ± 0.005	0.026 ± 0.008	1.319 ± 0.060	0.475 ± 0.015	0.331 ± 0.021	0.063 ± 0.014
	Swinging	0.051 ± 0.012	0.052 ± 0.009	1.414 ± 0.011	1.227 ± 0.028	0.453 ± 0.022	0.119 ± 0.021
	Rotating	0.057 ± 0.027	0.050 ± 0.009	1.102 ± 0.052	1.166 ± 0.041	0.337 ± 0.019	0.109 ± 0.014
2	Squatting	0.055 ± 0.008	0.054 ± 0.010	1.016 ± 0.040	0.951 ± 0.033	0.163 ± 0.007	0.131 ± 0.030
	Swinging	0.067 ± 0.012	0.052 ± 0.011	1.115 ± 0.023	1.214 ± 0.016	0.112 ± 0.016	0.047 ± 0.024
	Rotating	0.065 ± 0.023	0.043 ± 0.003	1.283 ± 0.039	0.793 ± 0.035	0.177 ± 0.050	0.051 ± 0.016
3	Squatting	0.087 ± 0.012	0.039 ± 0.010	1.217 ± 0.066	0.910 ± 0.043	0.085 ± 0.012	0.041 ± 0.022
	Swinging	0.104 ± 0.016	0.058 ± 0.011	1.138 ± 0.486	1.296 ± 0.092	0.100 ± 0.019	0.064 ± 0.021
	Rotating	0.069 ± 0.015	0.021 ± 0.216	1.290 ± 0.052	1.218 ± 0.117	0.097 ± 0.021	0.061 ± 0.006
4	Squatting	0.105 ± 0.036	0.024 ± 0.005	1.314 ± 0.042	1.317 ± 0.044	0.086 ± 0.020	0.044 ± 0.008
	Swinging	0.034 ± 0.009	0.032 ± 0.005	1.286 ± 0.031	0.801 ± 0.632	0.103 ± 0.008	0. 093 ± 0.015
	Rotating	0.034 ± 0.008	0.031 ± 0.015	1.213 ± 0.041	1.316 ± 0.097	0.069 ± 0.014	0.069 ± 0.017
5	Squatting	0.080 ± 0.017	0.076 ± 0.009	1.172 ± 0.041	1.025 ± 0.028	0.237 ± 0.018	0.225 ± 0.025
	Swinging	0.076 ± 0.017	0.068 ± 0.022	1.309 ± 0.037	0.076 ± 0.025	0.144 ± 0.012	0.110 ± 0.025
	Rotating	0.066 ± 0.026	0.044 ± 0.013	0.841 ± 0.034	0.852 ± 0.019	0.075 ± 0.022	0.080 ± 0.038

## Data Availability

The data are not publicly available. The data presented in this study are available on request from the first author and the corresponding author.

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
