# Peer review of "Design, Characterization, and Performance of Woven Fabric Electrodes for Electrocardiogram Signal Monitoring"

_sensors, 2022, doi:10.3390/s22155472_

Round 1
Reviewer 1 Report
This paper describes innovative work in developing woven textile electrodes, but there are very few details provided about the development of the electrodes, test protocol (including the number of trials run), and the quantification of performance of these electrodes when collecting the ECG signal. There is promising work here, but it needs to be presented far more clearly and explained thoroughly before this can be evaluated effectively.
Specific comments:
Lines 43-45: This statement is a bit off the mark. Integrating into clothing doesn't remove the need for conductive gel and adhesives. The need is still there, but it must be dealt with by testing garment fit and skin contact.
Line 57: Meant to say "satin" here?
Line 60: This systematic study investigates a sateen weave, among others: Arquilla, K.; Devendorf, L.; Webb, A.K.; Anderson, A.P. Detection of the Complete ECG Waveform with Woven Textile Electrodes. Biosensors 2021, 11, 331. https://doi.org/10.3390/bios11090331
Line 69: Do you mean filament or thread here?
Table 1: What does "Optimized Diameter" mean?
Lines 121-124: How were these three motions chosen?
Line 128: Be more explicit with the measurements you're using here for "signal" and "noise"
Line 152-154: What is the sodium chloride solution mentioned here? This was not described in the fabrication section of the paper, so it's difficult to put this into context.
Line 175: Was this confirmed experimentally over multiple trials?
Line 185: Please quantify "some noises"
Line 190-191: Why does sampling frequency impact AC impedance? This doesn't seem to be a differentiator between the electrode patterns.
Figure 4: Part C should be shown much larger to actually see the signals for comparison. At this size, no differences can be discerned.
Lines 203-204 and throughout: It is stated repeatedly that the different parts of the ECG waveform could be seen clearly. Was this measured in any way or reported quantitatively in any way? For example, how many anticipated heartbeat cycles are observable in each sample.
Figure 5: The caption should explain the key takeaways from each plot.
Line 216: Should say "positively related"
Line 237: Can the statement of "good signal stability and almost no noise" be quantified in some way?
Line 238 and throughout: are the electrodes truly optimized? What does this really mean?
Line 242: Please quantify "slight drift". In general, the assessments of signal quality are purely qualitative. There are standard metrics to quantify this that should be used here.
Figures 6 and 7: The "optimized" and "medical" designations are misleading. Consider labeling "woven textile" instead of "optimized"
General comments:
Would like to see an image of the fabric electrodes themselves, instead of just the modeled image in Figure 1. It is difficult to understand the structure of the circular woven electrodes without any images showing them. The edges must not be smooth?
Lots of information missing in the experimental methods. How many participants did you test with these electrodes? What were their characteristics?
Need to provide some explanation of how the textile electrodes were mounted on the body. How was consistent skin pressure maintained?
It is not clear that these electrodes were tested with more than one participant, which makes it very difficult to speak to their efficacy and reliability.
Reviewer 3 Report
1. The abstract could be copy-edited to enhance legibility.
Introduction:
2. You did not mention Holter ECG recorders that are more suitable for long-term ECG registration than patient monitors. These devices also require connections to electrodes attached to the subject's skin.
3. Lines 37-38: “However, long-term use on the human skin will cause skin allergies and other reactions” may be rephrased as “However, long-term use on the human skin may cause skin allergies and other reactions”.
4. Lines 48-49: “can have an impact on” may be rephrased as “may affect”.
Materials and Methods, section “Electrochemistry and human subject testing”:
5. There is no description of a human subject who participated in the testing procedure.
Results and Discussion:
6. Figure 2: The graphs could be enlarged and rearranged to a vertical orientation; the symbol “d” seems undefined. Do you mean “diameter”?
7. Figure 4: The graphs could be enlarged and rearranged to a vertical orientation. Moreover, the caption and graphs are split into two pages.
8. Figure 5: The graphs could be enlarged and rearranged to a vertical orientation; the symbol “Pw” seems undefined. Do you weft density in peaks/cm?
9. Line 153: “low enough to be ignored.” could be rephrased by “negligible”.
Minor points:
10. You can put a space between the last word and the reference(s).
Round 2
Reviewer 1 Report
Thanks for making these changes.
It is still not clear how or why the sodium chloride solution was used to measure the electrochemical properties. Which electrochemical properties were measured?
There is no Institutional Review Board mentioned that would have approved the study with human participants, which is an ethical concern.
It looks like the electrodes were woven in square form and then cut into circles. What was the motivation for this? What sealed the electrodes to prevent them from unraveling?
